**Data Availability Statement:** All relevant data are within the manuscript and its Supporting information files.

# Pathological complete response of adding targeted therapy to neoadjuvant chemotherapy for inflammatory breast cancer: A systematic review

Sudpreeda Chainitikun[1¤a], Jose Rodrigo Espinosa Fernandez[1¤b], James P. Long[2], Toshiaki Iwase[1], Kumiko Kida[1], Xiaoping Wang[1], Sadia Saleem[1], Bora Lim[1], Vicente Valero[1], Naoto T. Ueno[1]*

**1** Morgan Welch Inflammatory Breast Cancer Research Program and Clinic and Department of Breast Medical Oncology, The University of Texas MD Anderson Cancer Center, Houston, Texas, United States of America, **2** Department of Biostatistics, The University of Texas MD Anderson Cancer Center, Houston, Texas, United States of America

¤a Current address: Department of Medical Oncology, MedPark Hospital, Bangkok, Thailand
¤b Current address: Department of Medical Oncology, Instituto Nacional de Cancerologia INCAN, Mexico City, Mexico
* nueno@mdanderson.org

## Abstract

### Background

The current use of targeted therapy plus neoadjuvant chemotherapy for inflammatory breast cancer (IBC) is based on data extrapolated from studies in non-IBC. We conducted a systematic review to determine whether neoadjuvant chemotherapy plus targeted therapy results in a higher pathologic complete response (pCR) rate than neoadjuvant chemotherapy alone in patients with IBC.

### Method and findings

This systematic review was registered in the PROSPERO register with registration number CRD42018089465. We searched MEDLINE & PubMed, EMBASE, and EBSCO from December 1998 through July 2020. All English-language clinical studies, both randomized and non-randomized, that evaluated neoadjuvant systemic treatment with or without targeted therapy before definitive surgery and reported the pCR results of IBC patients. First reviewer extracted data and assessed the risk of bias using the Risk of Bias In Non-randomized Studies of Interventions tool. Second reviewer confirmed the accuracy. Studies were divided into 3 groups according to systemic treatment: chemotherapy with targeted therapy, chemotherapy alone, and high-dose chemotherapy with hematopoietic stem cell support (HSCS). Of 995 screened studies, 23 with 1,269 IBC patients met the inclusion criteria. For each of the 3 groups of studies, we computed a weighted average of the pCR rates across all studies with confidence interval (CI). The weighted averages (95% CIs) were as follows: chemotherapy with targeted therapy, 31.6% (26.4%-37.3%), chemotherapy alone, 13.0% (10.3%-16.2%), and high-dose chemotherapy with HSCS, 23.0% (18.7%-27.7%). The high

**Funding:** This research was supported by the Morgan Welch Inflammatory Breast Cancer Research Program, a State of Texas Rare and Aggressive Breast Cancer Research Program Grant, and National Institutes of Health/National Cancer Institute grant P30 CA016672 (Cancer Center Support Grant; used the Biostatistics Resource Group and the Clinical and Translational Research Center) and the Center for Clinical and Translational Sciences (CCTS UL1TR000371). The funders had no role in study design, data collection and analysis, decision to publish, or preparation of the manuscript.

**Competing interests:** The authors have declared that no competing interests exist.

pCR by targeted therapy group came from anti-HER2 therapy, 54.4% (44.3%-64.0%). Key limitations of this study included no randomized clinical studies that included only IBC patients.

## Conclusion

Neoadjuvant chemotherapy plus targeted therapy is more effective than neoadjuvant chemotherapy alone for IBC patients. These findings support current IBC standard practice in particular the use of anti-HER2 targeted therapy.

## Introduction

Inflammatory breast cancer (IBC) is the most lethal and aggressive form of breast cancer and is associated with worse survival outcomes and prognosis than non-IBC. IBC is also a rare disease, accounting for 1% to 5% of all breast cancer cases [1, 2]. Clinically, IBC is defined as diffuse erythema and edema (peau d'orange) involving approximately one-third or more of the breast skin with or without an underlying palpable mass. This definition was introduced in the eighth edition of the American Joint Committee on Cancer (AJCC) [3]. Importantly, IBC is primarily a clinical diagnosis [3].

Management of IBC with tri-modality treatment (chemotherapy, surgery, and radiation therapy) can increase survival [4]. The recommended approach in patients with newly diagnosed IBC is neoadjuvant systemic treatment, including chemotherapy with or without targeted therapy, followed by definitive surgery and then radiation therapy [5, 6].

Over the past decade, many novel targeted therapies have become standard treatment of breast cancer. Unlike chemotherapy, targeted therapy directly affects cancer cells and largely spares normal cells, which can improve efficacy and reduce toxicity. Among the targeted therapies used against breast cancer are trastuzumab, lapatinib, TDM-1, and pertuzumab, which target human epidermal growth factor receptor 2 (HER2) [7]; bevacizumab, which blocks vascular epidermal growth factor receptor (anti-VEGF) and results in reduced tumor angiogenesis [8]; and panitumumab, which blocks epidermal growth factor receptor (anti-EGFR) and inhibits tumor growth by inhibition of cell survival pathways [9]. All of these targeted therapies have been shown in clinical trials to benefit breast cancer patients, but the trials have been focused on patients with non-IBC.

Many clinical trials have shown that adding targeted therapy to neoadjuvant chemotherapy improves the pathological complete response (pCR) rate for breast cancer [10–14]. pCR in breast cancer, defined as a lack of invasive cancer cells in surgical specimens of both the breast and axillary lymph nodes after neoadjuvant treatment, is associated with improved long-term survival [15–17]. The U.S. Food and Drug Administration has endorsed pCR as a surrogate endpoint for overall survival in breast cancer [18].

Because of the rarity of IBC, none of the randomized clinical trials evaluating the efficacy of adding targeted therapy to neoadjuvant chemotherapy have been limited to IBC patients. Furthermore, most of the clinical trials have enrolled mixed populations of IBC and non-IBC patients. However, at present, even though all standard guidelines define IBC separately from non-IBC, IBC patients are treated with the same combinations of targeted therapy and neoadjuvant chemotherapy used for non-IBC patients. Systemic treatment for IBC is based on data extrapolated from non-IBC clinical trials. We lack strong evidence to support a benefit of these systemic treatment regimens, especially those including targeted therapy, for IBC patients.

Some clinical studies have shown that the same systemic treatment was less effective for IBC compared to non-IBC [19, 20]. The best way to clarify the benefit of adding targeted therapy to neoadjuvant chemotherapy for IBC, is by conducting a systematic review to extract the IBC data from clinical trials.

## Methods

This systematic review was conducted in accordance with the Preferred Reporting Items for Systematic Reviews and Meta-Analyses statement [21, 22]. This study was registered in the PROSPERO register with registration number CRD42018089465 [23]. The key review question was, Has addition of targeted therapies for breast cancer improved pCR rates of IBC patients compared with pCR rates with neoadjuvant chemotherapy alone?

### Data sources and searches

We systematically searched the MEDLINE & PubMed, EMBASE, and EBSCO databases using the following search terms: "inflammat*" and "breast". The Medical Subject Heading term "inflammatory breast neoplasms" was applied and adapted for each database as necessary. The original protocol in PROSPERO was registered in 2018 and purposed to update the articles in the last 20 years [23]. Searches were limited to clinical trials during 1998–2020 (last search update was performed on July 15, 2020). The search strategy also included hand-searching the reference lists of relevant articles. To be included, studies had to be published in English. All searches were guided by an expert team from the Research Medical Library at The University of Texas MD Anderson Cancer Center. The search strategy for Ovid MEDLINE is provided in S1 Table.

### Study selection

To be included in this review, studies had to meet every one of the criteria listed below.

- Participants: The study included IBC patients of any age and with any breast cancer molecular subtype for whom the aim was definitive surgery to remove the primary tumor. Studies with both IBC and non-IBC patients were eligible if the outcome for only IBC patients could be extracted.

- Interventions/Comparators: The study evaluated neoadjuvant systemic treatment with targeted therapy (Interventions), or without targeted therapy (Comparators) before definitive surgery.

- Outcome measure: The outcome measure was pCR rate defined as the proportion of patients with absence of invasive cancer cells from breast or breast and axillary lymph nodes after neoadjuvant systemic treatment.

- Study type: The study was a clinical study, randomized or nonrandomized, retrospective or prospective, and reported the outcomes of IBC patients. Registry database studies with unclear treatment strategy, abstracts without full articles, and case reports were excluded. The number of excluded studies (including reasons for exclusion) was recorded at each stage.

After removal of duplicate studies, all titles and abstracts were screened by independent reviewers (SC, JE). When initial screening indicated that a study was potentially eligible for inclusion, the full text was assessed to determine eligibility. Any discrepancies during article screening and full-text assessment were resolved by an independent third reviewer (NU). In

the case of multiple publications describing the same population, only the most recent publication or relevant information was included.

### Data extraction and quality assessment

For each study selected for inclusion, a first reviewer extracted relevant data from the full-text article, and a second reviewer confirmed the accuracy of the first reviewer's work. Two authors, SC and JE, performed this work, with SC serving as first and JE serving as second reviewer for half of the articles and JE serving as first and SC serving as second reviewer for the other half of the articles. The following data were extracted: publication details (authors, phase of clinical trial, and year(s) of accrual), sample size (IBC and non-IBC), baseline patient and disease characteristics (age, stage, estrogen receptor status, progesterone receptor status, HER2 status), treatment regimen, and pCR rate.

The risk of bias for each study was assessed by using the Risk of Bias In Nonrandomized Studies of Interventions (ROBIN-I) tool [24]. ROBINS-I covers 7 domains through which bias might be introduced, including bias due to confounding, selection bias, classification of interventions, deviations, missing data, outcome measurement, and selection of the reported results. For each study, 2 reviewers (SC, JE) described the risk of bias in each domain as low, moderate, serious, critical, or no information available and described the overall risk of bias using the same scale. Any discrepancies were resolved by the third reviewer (NU).

### Data synthesis and analysis

The eligible studies were divided into 3 groups according to systemic treatment: chemotherapy with targeted therapy, chemotherapy alone, and high-dose chemotherapy, defined as chemotherapy requiring hematopoietic stem cell support (HSCS) regardless of chemotherapy dosage. Chemotherapy administered with granulocyte colony-stimulating factor (or another agent in the same group) without HSCS was not classified as high-dose chemotherapy. Currently, the high-dose chemotherapy is not a standard treatment for IBC, we separately reported data in S1 Appendix.

For each group, we calculated a weighted-average pCR rate, defined as the total number of patients achieving pCR across all studies in the group divided by the total number of patients across all studies in the group. Confidence intervals (CIs) for proportions were computed using the method of Clopper and Pearson.

## Results

After duplicates were removed, titles and abstracts from 995 articles were screened, and 888 articles were excluded. A total of 107 full-text articles were reviewed, and 84 articles were excluded. Evidence search and study selection are summarized in Fig 1. Of the 23 studies included, 15 included only IBC patients, and 8 included both IBC and non-IBC patients. Of these 6 studies [25–30] of high-dose chemotherapy were separately reported in S1 Appendix. We identified no randomized clinical trial to determine the efficacy of adding targeted therapy to chemotherapy restricted to IBC patients.

### Study characteristics

Characteristics of the 23 studies and the patients in those studies are summarized in Table 1 and S3 Table. The studies included 1269 IBC patients, of whom 329 received chemotherapy with targeted therapy, 571 received chemotherapy alone, and 369 received high-dose chemotherapy (S1 Appendix). Only 1 study compared chemotherapy with targeted therapy versus

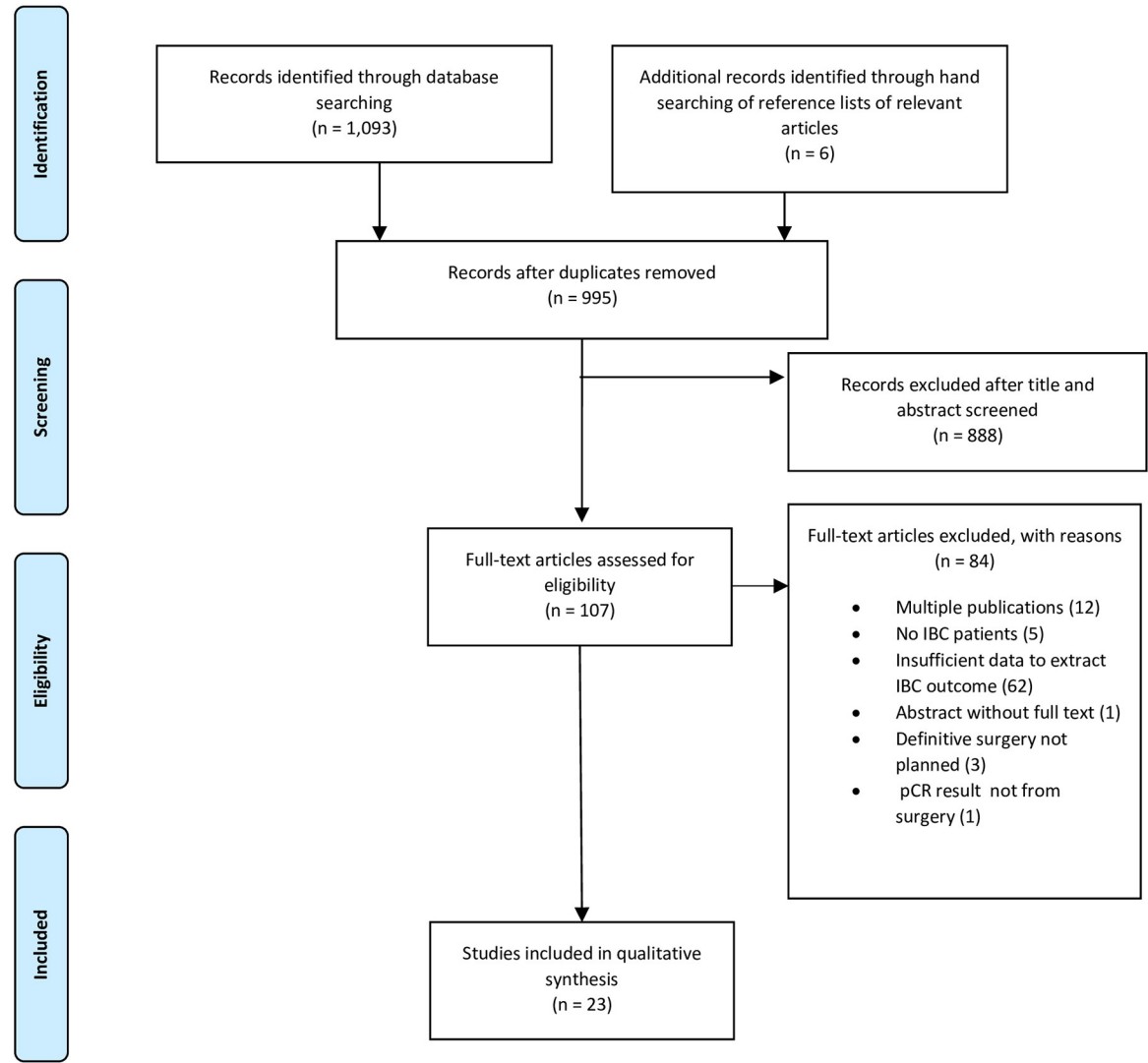

**Fig 1. Evidence search and selection.**

chemotherapy alone [31]. That study, by Nahleh et al [31], was a randomized study with both IBC and non-IBC patients, and the authors reported the pCR rates of IBC patients in each arm. Therefore, results from this study are reported in both the chemotherapy with targeted therapy and chemotherapy alone parts of Table 1. Three studies enrolled patients with metastatic disease for whom definitive surgery (mastectomy) was planned [32–34], and 1 study enrolled patients with locally recurrent cancer after breast-conserving surgery who were eligible for surgery [35]. The pCR rates for IBC patients and non-IBC patients (if applicable) in each study are shown in Table 2.

## Chemotherapy with targeted therapy

Our review included 9 studies of chemotherapy with targeted therapy that included 329 IBC patients. Of these 9 studies, only the above-mentioned study by Nahleh et al [31] was a randomized study; the others were single-arm studies. The studies accrued patients during 2005–2015. Patients in these studies received 4 types of targeted therapy: anti-HER2 (trastuzumab,

**Table 1. Characteristics of the included studies and baseline characteristics of the patients in those studies.**

| First author/ Reference | Phase | Inclusion criteria | Total (No.) | HER2+ (No.) | ER+ (No.) | PR+ (No.) | HR+ (No.) | IBC (No.) | Median age (range), y | Years of accrual | Risk of bias |
|---|---|---|---|---|---|---|---|---|---|---|---|
| **Chemotherapy with targeted therapy** | | | | | | | | | | | |
| Pizzuti [37] | 2 | II, III, HER2+ | 45 | 45 | 23 | 17 | 24 | 9 | 45 (32–69) | 2008–2014 | Low |
| Torrisi [34] | 2 | III, IV, HER2+ | 32 | 32 | 16 | 11 | NR | 13 | 47 (28–69) | 2007–2008 | Low |
| Palazzo [35] | 2 | III, local recurrence | 34§ | 14 | NR | NR | 18 | 34 | 48 (26–67) | 2010–2013 | Moderate |
| Pierga [36] | 2 | III, HER2+ | 52 | 52 | 18 | 9 | NR | 52 | 52.2 (44–60) | 2008–2009 | Low |
| Nahleh* [31] | 2 | II, III, HER2- | 98 | 0 | NR | NR | 144 | 10 | 51.7 (22–71) | 2010–2012 | Moderate |
| Bertucci [33] | 2 | III, HER2- | 100 | 1‡ | 42 | 37 | 45 | 100 | 49 (42–47) | 2009–2010 | Moderate |
| Matsuda [32] | 2 | III, IV, HER2- | 40 | 0 | NR | NR | 21 | 40 | 57 (23–68) | 2010–2015 | Low |
| Boussen [39] | 2 | III, IV | 49 | 32 | NR | NR | NR | 49 | 53.4 (30–74) | 2005–2006 | Moderate |
| Andreopoulou [38] | 1/2 | II, III, HER2- | 55 | 0 | 12 | 9 | 13 | 22 | 54.5 (34–77) | 2007–2011 | Serious |
| **Chemotherapy alone** | | | | | | | | | | | |
| Cristofanilli [45] | 2 | III | 44 | NR | 23 | NR | NR | 44 | 51 (27–28) | 1994–1998 | Serious |
| de Matteis [46] | 2 | II, III | 30 | NR | NR | NR | NR | 9 | 48 (28–68) | 1999–2000 | Moderate |
| Ditsch [41] | 3 | III | 101 | NR | NR | NR | 20 | 101 | 53 (33–64) | 1998–2002 | Low |
| Baldini [44] | 2 | III | 68 | NR | 15 | 10 | NR | 68 | 50 (30–70) | 1985–1997 | Moderate |
| Veyret [42] | 2 | III | 120 | NR | NR | NR | NR | 120 | NR | 1990–1992 | Low |
| Kummel [43] | 2 | III | 34 | NR | 2 | NR | NR | 7 | 56 (36–73) | 1996–1998 | Low |
| Costa [40] | 3 | III | 287† | 93 | 172 | 119 | 181 | 93 | 53 (29–78) | 2002–2005 | Serious |
| Ellis [20] | 3 | II, III | 372 | 92 | NR | NR | 185 | 115 | 52 (22–77) | 2001–2005 | Low |
| Nahleh* [31] | 2 | II, III/ HER2- | 113 | 0 | NR | NR | 144 | 14 | 51.3 (31–75) | 2010–2012 | Moderate |

Abbreviations: ER+, estrogen receptor positive; HER2+, HER2 positive; HR+, hormone receptor positive; IBC, inflammatory breast cancer; NR, not reported; PR+, progesterone receptor positive

*Nahleh et al. randomized patients to chemotherapy with bevacizumab (n = 98) or chemotherapy without bevacizumab (n = 113).

†Not including operable breast cancer patients.

‡One patient with HER2+ disease was enrolled by mistake and included in the analysis (intention-to-treat).

§All of 34 patients were stage III IBC.

**Table 2. pCR rates in each study.**

| First author/ reference | IBC patients (No.) | Non-IBC patients (No.) | Regimen | pCR rate, % IBC | pCR rate, % Non-IBC | pCR rate, % Overall |
|---|---|---|---|---|---|---|
| **Chemotherapy with targeted therapy** | | | | | | |
| Pizzuti [37] | 9 | 36 | Neoadj. (docetaxel → epirubicin/cyclophosphamide) + **trastuzumab** | 66.70 | NR | 62.2 |
| Torrisi [34] | 13 | 19 | Neoadj. Pegylated liposomal doxorubicin/cisplatin/5-FU/ **trastuzumab** | 54 | NR | 41 |
| Palazzo [35] | 34 | 0 | Neoadj. Paclitaxel/carboplatin/cyclophosphamide/ **bevacizumab,** If HER2 + add **trastuzumab,** If HR+ add letrozole | 29 / 57 (HER2+) | NR | NR |
| Pierga [36] | 52 | 0 | Neoadj. FEC/**bevacizumab** → docetaxel/**bevacizumab**/**trastuzumab** → surgery | 63.50 | NR | NR |
| Nahleh [31] | 10 | 88 | Neoadj. Arm 1. Nab-paclitaxel/**bevacizumab** → dose-dense AC→ surgery | 30 | 36.40 | 35.70 |
| Bertucci [33] | 100 | 0 | Neoadj. (FEC → docetaxel) + **bevacizumab** → surgery | 19 | NR | NR |
| Matsuda [32] | 40 | 0 | Neoadj. Carboplatin/nab-paclitaxel/**panitumumab** → FEC→ surgery | 28 / 42 (TNBC) | NR | NR |
| Boussen [39] | 49 | 0 | Neoadj. Paclitaxel/**lapatinib** → surgery | 18.20 | NR | NR |
| Andreopoulou [38] | 22 | 33 | Neoadj. (paclitaxel → AC) + **tipifarnib** → surgery | 4 | 15 | NR |
| **Chemotherapy alone** | | | | | | |
| Cristofanilli [45] | 44 | 0 | Neoadj. FAC (add paclitaxel if no response) → surgery | 14 | NR | NR |
| de Matteis [46] | 9 | 21 | Neoadj. Epirubicin/docetaxel → surgery | 11.10 | NR | 13.30 |
| Ditsch [41] | 101 | 0 | Neoadj. Arm 1. Dose-dense: epirubicin/paclitaxel → surgery Arm 2. Standard: epirubicin/paclitaxel → surgery | 11 | NR | NR |
| Baldini [44] | 68 | 0 | Neoadj. FEC or FAC → surgery | 5.90 | NR | NR |
| Veyret [42] | 120 | 0 | Neoadj. FEC +/- lenograstim → surgery | 14.70 | NR | NR |
| Kummel [43] | 7 | 27 | Neoadj. Dose-dense: epirubicin → dose-dense: docetaxel → surgery | 0 | 11.10 | 8.8 |
| Costa [40] | 93 | 194 | Neoadj. Docetaxel/doxorubicin/cyclophosphamide (add vinorelbine/ capecitabine if no response) → surgery | 8.60 | 11.3* | 10.5 |
| Ellis [20] | 115 | 249 | Neoadj. Arm 1. AC → paclitaxel → surgery Arm 2. Weekly doxorubicin/oral cyclophosphamide→ paclitaxel → surgery | 19.80 | 23.70 | 22.50 |
| Nahleh [31] | 14 | 99 | Neoadj. Arm 2. Nab-paclitaxel → dose-dense AC (or vice versa) → surgery | 14.30 | 22.20 | 21.20 |

Abbreviations: 5-FU, 5-fluorouracil; AC, doxorubicin and cyclophosphamide; FAC, 5-FU, doxorubicin, and cyclophosphamide; FEC, 5-FU, epirubicin, and cyclophosphamide; HER2+, HER2 positive; HR+, hormone receptor positive; neoadj., neoadjuvant therapy; NR, not reported; pCR, pathological complete response; TNBC, triple-negative breast cancer.

*pCR rate of locally advanced breast cancer patients (not including operable breast cancer).

lapatinib), anti-VEGF (bevacizumab), anti-EGFR (panitumumab), and RAS-targeting (tipifarnib).

Three studies [34, 36, 37] included only HER2-positive patients, and 4 studies [31–33, 38] included only HER2-negative patients. Boussen et al. [39] initially enrolled IBC patients with HER2-positive and HER2-negative disease; however, HER2-negative patients were excluded from the analysis because of lack of efficacy of lapatinib and slow accrual. Palazzo et al. [35] also enrolled IBC patients with HER2-positive and HER2-negative disease and separately reported the pCR rate for the HER2-positive group.

Among the studies of chemotherapy with targeted therapy, the highest reported pCR rate was 66.7%, reported for the HER2-positive IBC patients treated with trastuzumab in the

study by Pizzuti et al. [37]. That study included only 9 IBC patients; however, high pCR rates (57.0% and 63.5%, respectively) were also found in 2 larger studies that enrolled only IBC patients and treated them with trastuzumab plus bevacizumab [35, 36]. In contrast, lapatinib combined with chemotherapy in HER2-positive IBC patients produced a pCR rate of only 18.2% [39].

In 4 studies of chemotherapy plus targeted therapy, patients were treated with bevacizumab: the studies by Palazzo et al. [35] and Pierga et al. [36] of bevacizumab plus trastuzumab, mentioned in the preceding paragraph, and studies by Bertucci et al. and Nahleh et al. Pierga et al. [36] enrolled only HER2-positive IBC patients. Palazzo et al. [35] enrolled HER2-positive (n = 14) and HER2-negative (n = 20) IBC patients and treated them with chemotherapy with bevacizumab, with trastuzumab added for HER2-positive disease. The pCR rate for all patients was 29.0% (10 of 34 patients achieved a pCR). The study by Bertucci et al. [33] included the largest number of IBC patients (n = 100) and had a pCR rate of 19.0% after chemotherapy plus bevacizumab. In the study by Nahleh et al. [31], the randomized clinical trial, chemotherapy plus bevacizumab produced a higher pCR rate than did chemotherapy alone among IBC patients (n = 24 in the 2 treatment groups combined), but the difference was not statistically significant owing to the small sample size (30.0% vs 14.3%, p = 0.61).

In the Matsuda et al. study [32], HER2-negative IBC patients were treated with chemotherapy plus panitumumab, and the pCR rate was 28.0% (and 42.0% in the subgroup with triple-negative IBC).

Chemotherapy plus tipifarnib, which targets the RAS pathway, produced a pCR rate of only 4.0% in HER2-negative IBC patients [38].

## Chemotherapy alone

Our review included 9 studies of chemotherapy alone that included 571 IBC patients: 5 randomized studies [20, 31, 40–42] and 4 single-arm studies [43–46]. The 6 studies that started enrollment before 2001, when a pivotal trial of anti-HER2 therapy was published [47], did not report the number of patients with HER2 overexpression because HER2 testing was not available.

In all 9 studies, IBC patients were treated with anthracycline-based chemotherapy (doxorubicin or epirubicin). The pCR rates ranged from 0% to 20.1%. The lowest pCR rate was observed in a study by Kummel et al. [43], who evaluated dose-dense epirubicin followed by dose-dense docetaxel; this study included only 7 IBC patients. The highest pCR rate was observed in a study by Ellis et al. [20] of doxorubicin plus cyclophosphamide followed by paclitaxel, similar to the current standard regimen. Four studies [20, 31, 40, 43] also reported the pCR rate in non-IBC patients. Notably, all of those studies showed higher pCR rates for non-IBC than for IBC.

## Weighted averages pCR rates

The pCR rates with 95% CIs for the IBC patients in each study and the weighted-average pCR rates for each group are shown in Fig 2. The weighted-average pCR rates were 31.6% (95% CI, 26.4%-37.3%) for chemotherapy with targeted therapy, 13% (95% CI, 10.3%-16.2%) for chemotherapy alone, and 23% (95% CI, 18.7%-27.7%) for high-dose chemotherapy (S1 Fig).

In the subgroup of 5 studies [34–37, 39] in which 105 HER2-positive IBC patients were treated with chemotherapy and either trastuzumab or lapatinib, 57 patients achieved a pCR, and the weighted-average pCR rate was numerical highest of 54.3% (95% CI, 44.3%-64.0%).

patients in each group

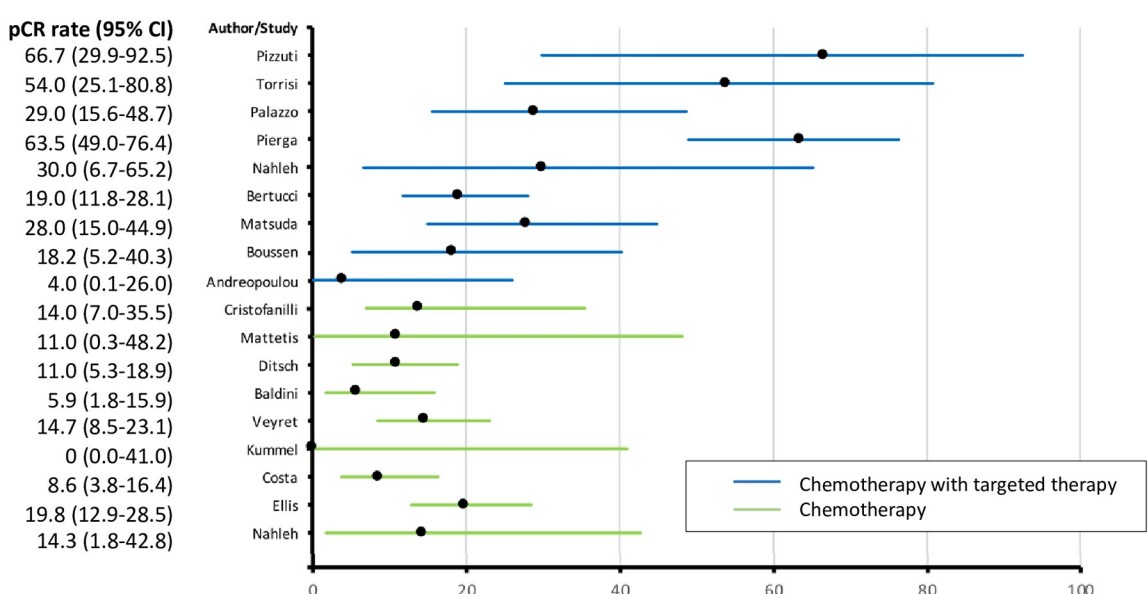

**pCR rate (95% CI)**

| | |
|---|---|
| 66.7 (29.9-92.5) | Pizzuti |
| 54.0 (25.1-80.8) | Torrisi |
| 29.0 (15.6-48.7) | Palazzo |
| 63.5 (49.0-76.4) | Pierga |
| 30.0 (6.7-65.2) | Nahleh |
| 19.0 (11.8-28.1) | Bertucci |
| 28.0 (15.0-44.9) | Matsuda |
| 18.2 (5.2-40.3) | Boussen |
| 4.0 (0.1-26.0) | Andreopoulou |
| 14.0 (7.0-35.5) | Cristofanilli |
| 11.0 (0.3-48.2) | Mattetis |
| 11.0 (5.3-18.9) | Ditsch |
| 5.9 (1.8-15.9) | Baldini |
| 14.7 (8.5-23.1) | Veyret |
| 0 (0.0-41.0) | Kummel |
| 8.6 (3.8-16.4) | Costa |
| 19.8 (12.9-28.5) | Ellis |
| 14.3 (1.8-42.8) | Nahleh |

| Regimen | Number of IBC patients/studies | pCR rate (95% CI) | pCR definition (Absence of invasive tumor cells) |
|---|---|---|---|
| 1. Chemotherapy with targeted therapy | 329/9 | 31.6 (26.4-37.3) | Breast and axillary lymph nodes |
| – Chemotherapy with anti-HER2 targeted therapy | 157/5 | 54.3 (44.3-64.0) | Breast and axillary lymph nodes |
| – Chemotherapy with non-anti-HER2 targeted therapy | 172/4 | 20.2 (14.4-27.1) | Breast and axillary lymph nodes |
| 2. Chemotherapy alone | 571/9 | 13 (10.3-16.2) | Breast and axillary lymph nodes |

**Fig 2. pCR rates for IBC patients in each study and weighted-average pCR rates for IBC patients in each group.**

### Risk of bias

On assessment for risk of bias using the ROBINS-I tool, 4 studies [25, 38, 40, 45] were rated as having a serious risk of bias overall (Table 1). Two studies were in the chemotherapy alone group, 1 study was in the chemotherapy with targeted therapy group, and 1 study was in the high-dose chemotherapy group. The remaining studies were rated as having a moderate or low risk of bias overall. The ratings for the domains contributing to the overall risk of bias are shown in S2 Table.

### Discussion

In this systematic review, we found that the addition of targeted therapy to neoadjuvant chemotherapy for IBC patients resulted in a higher weighted-average pCR rate. There was a difference of 18.6 percentage points between the weighted-average pCR for chemotherapy alone and the weighted-average pCR for targeted therapy plus chemotherapy by all IBC population

regardless subtypes. Anti-HER2 targeted therapy showed the promising results for HER2-positive IBC patients with the weighted-average pCR rate of 54.3%.

A previous systematic review evaluated the relationship between the dose intensity of chemotherapy and response and survival in IBC patients [48]. That review did not include targeted therapy because targeted therapy was not available during the period covered. However, the pCR rate reported for anthracycline-based neoadjuvant chemotherapy, approximately 11% to 14%, is similar to the weighted-average pCR rate of 13% (95% CI, 10.3%-16.2%) that we found for the chemotherapy-alone group in our current study. It is interesting that despite extensive changes in breast cancer treatment in recent years, comparison of the findings from the previous review and our current review suggests that the efficacy of chemotherapy alone for IBC has not improved.

At present, various targeted therapies have been established in breast cancer treatment worldwide. Particular, the addition of anti-HER2 targeted therapy to chemotherapy has showed an improvement of the efficacy for breast cancer treatment [7]. Our results from this systematic review support these findings; specifically, subgroup analysis showed the highest weighted-average pCR rate (54.3%, 95% CI, 44.3%-64.0%) for anti-HER2 therapy (trastuzumab and lapatinib) in the HER2-positive IBC subgroup, and a weighted-average pCR rate of only 20.2% (95% CI 14.4%-27.1%) for other targeted therapies (bevacizumab, panitumumab, and tipifarnib) in IBC patients. Taken together, these findings imply that the high pCR rate with addition of targeted therapy to chemotherapy was due mainly to anti-HER2 targeted therapy. Careful interpretation of our findings is needed, however, because the pCR rate for the chemotherapy alone group was calculated from results in IBC patients with all breast cancer subtypes, whereas the pCR rate for anti-HER2 plus chemotherapy was calculated from results specifically in patients with HER2-positive IBC. The NOAH trial [14, 49], randomized controlled trial for HER2-positive locally advanced breast cancer or IBC, also reported that the pCR significantly related with longer overall survival in patient treated with trastuzumab. However, even if the NOAH trial enrolled 61 IBC patients but did not meet the criteria of this systematic review.

For anti-VEGF therapy (bevacizumab), effective predictive markers have not been established. Interestingly, in this review, we found that in 2 studies [35, 36], the combination of trastuzumab plus bevacizumab and chemotherapy produced very high pCR rates in HER2-positive IBC patients (57.0% and 63.5%, respectively), but these pCR rates were not higher than those in studies in which HER2-positive IBC patients were treated with single-agent trastuzumab and chemotherapy (66.7% and 54.0%, respectively) [34, 37]. These results imply no synergy between bevacizumab and trastuzumab and suggest that the high pCR rate in the dual-targeted-therapy studies was produced by trastuzumab and not bevacizumab. However, cross clinical trial comparison needs careful interpretation. Similarly, results from 2 phase 3 randomized studies, the ARTemis [50] and Gepaquinto [51] studies, showed no significant increase in the pCR rate with addition of bevacizumab to neoadjuvant chemotherapy in subgroup analysis of patients with IBC (or T4 tumors). In current practice, there is no role for bevacizumab for breast cancer.

We also calculated weighted-average pCR rates for non-IBC patients across studies in treatment groups. This analysis included the 4 studies in the chemotherapy-with-targeted-therapy group and the 4 studies in the chemotherapy-alone group that included non-IBC patients. We found that weighted-average pCR rates for non-IBC were higher than those for IBC (chemotherapy plus targeted therapy: 42.7% [95% CI, 34.6%-51.0%] vs 31.6% [95% CI, 26.4%-37.3%]; chemotherapy alone: 15.6% [95% CI, 11.8%-20.0%] vs 13.0% [95% CI, 10.3%-16.2%]). In our systematic review, results for non-IBC patients cannot be directly compared to results for IBC patients because the search strategy was initiated on a research question that focused on IBC

patients, and the study was not designed for non-IBC patients. Hence, the non-IBC results in our systematic review were highly selected, may not be a good representation for the overall non-IBC. However, as expected because of the aggressive nature of IBC, we observed that pCR rates for IBC were lower than those for non-IBC in the 2 groups in which studies included non-IBC patients.

In interpreting the results of our systematic review, it is important to consider that the definitions of IBC were not consistent across all the studies. Most of the studies included in our review used the current AJCC definition of IBC (T4d disease), but 4 studies [8, 26, 28, 30, 36] also enrolled IBC patients defined on the basis of the French Poussée Evolutive (PEV) breast cancer classification [52]. The PEV classification comprises 4 stages, of which PEV2 (inflammatory skin changes involving less than half of the breast surface) and PEV3 (inflammatory skin changes involving more than half of the breast surface) are classified as IBC [52]. Notably, IBC diagnosed according to the PEV2 classification would not be considered IBC according to the current AJCC definition if the inflammatory changes involved less than 30% of the breast surface. However, we found only 8 patients diagnosed with IBC according to the PEV2 criteria, in the Pierga et al. study [36], and 2 patients diagnosed with IBC on the basis of dermal lymphatic invasion without clinical signs, in the Sportes et al. study [28]. The reports for the remaining studies did not mention the number of patients with IBC diagnosed according to definitions other than the current AJCC definition. Given that our review most likely included only a very small number of patients who did not have IBC diagnosed by the current AJCC criteria, we expect that our final conclusions would be the same even if these patients were excluded.

The definition of pCR also was not consistent across all the studies included in our review; in 3 studies [27, 29, 30] in the high-dose chemotherapy group was defined as absence of tumor cells at the site of the primary breast tumor, with axillary lymph node findings not taken into account (S1 Appendix). This may have led to overestimation of the pCR rates in the high-dose chemotherapy group. But this would not change our principal finding that chemotherapy with targeted therapy was associated with the highest pCR rate. Furthermore, high-dose chemotherapy is not a current standard treatment for IBC owing to concerns regarding toxicity.

Our study also has other limitations. Even though the chemotherapy regimens used in the studies we reviewed were anthracycline- and/or taxane-containing regimens, similar to what is currently used in clinical practice, we did not analyze the dose-intensity or the schedule of chemotherapy including chemotherapy combined with targeted therapy. Dosage and schedule differed between studies. We minimized this bias by separating the high-dose chemotherapy group from the chemotherapy-alone group. Also, as previously mentioned, high-dose chemotherapy is not a standard treatment for IBC at present. Another limitation is that our review did not include newer targeted therapy (e.g., pertuzumab) or immunotherapy that has recently demonstrated efficacy in breast cancer because we found no reports of the results of such therapy in IBC patients [10, 53]. The last limitation we note is that our review did not include any randomized studies evaluating the efficacy of adding targeted therapy to neoadjuvant chemotherapy limited to IBC patients. The absence of such studies is a consequence of the rarity of the disease, as previously mentioned.

To our knowledge, this is the first systematic review to evaluate the efficacy of adding targeted therapy to neoadjuvant chemotherapy specifically in IBC. At present, all guidelines still recommend targeted therapy as one part of systemic treatment for IBC, even though there has been a lack of strong evidence supporting this practice [54, 55]. The findings of our review can be interpreted as strong evidence confirming the benefit of adding targeted therapy, particularly anti-HER2 therapy, and fill this gap in data supporting IBC guidelines. It is challenging to find data supporting treatment recommendations for a rare disease like IBC. Most IBC clinical

trials face the problems of slow accrual and rapid change in standard treatment, which can lead to protocol amendment or trials being stopped. Another strength of our study is that we reviewed studies that collectively enrolled 1,269 IBC patients, the largest number to be analyzed for this rare disease.

## Conclusions

Our systematic review confirmed the efficacy of adding targeted therapy to neoadjuvant chemotherapy for IBC patients. This provides strong evidence to support the systemic treatment strategy in the current IBC guidelines. The data justifies the use of current anti-HER2 targeted therapy for patients with HER2-positive disease despite there are no randomized studies.

## Supporting information

**S1 Table. Search strategy Ovid MEDLINE (July 15, 2020).**
(DOCX)

**S2 Table. Risk of bias by ROBIN-I.**
(DOCX)

**S3 Table. Characteristics of the included studies and baseline characteristics of the patients in high-dose chemotherapy studies.**
(DOCX)

**S4 Table. pCR rates in each study of high-dose chemotherapy studies.**
(DOCX)

**S1 Checklist. PRISMA 2009 checklist.**
(DOC)

**S1 Fig. pCR rates for IBC patients in each study and weighted-average pCR rates for IBC patients of high-dose chemotherapy studies.**
(TIF)

**S1 Appendix. High-dose chemotherapy.**
(DOCX)

## Acknowledgments

We thank Stephanie P. Deming (Scientific Publications, Research Medical Library, MD Anderson Cancer Center) for help in editing this article.

## Author Contributions

**Conceptualization:** Sudpreeda Chainitikun, Jose Rodrigo Espinosa Fernandez, Toshiaki Iwase, Kumiko Kida, Xiaoping Wang, Sadia Saleem, Bora Lim, Vicente Valero, Naoto T. Ueno.

**Data curation:** Sudpreeda Chainitikun, Jose Rodrigo Espinosa Fernandez, James P. Long.

**Formal analysis:** Sudpreeda Chainitikun, Jose Rodrigo Espinosa Fernandez, James P. Long.

**Funding acquisition:** Naoto T. Ueno.

**Investigation:** Sudpreeda Chainitikun.

**Methodology:** Sudpreeda Chainitikun, Jose Rodrigo Espinosa Fernandez, James P. Long.

**Software:** James P. Long.

**Supervision:** Sudpreeda Chainitikun, Toshiaki Iwase, Kumiko Kida, Xiaoping Wang, Sadia Saleem, Bora Lim, Vicente Valero, Naoto T. Ueno.

**Writing – original draft:** Sudpreeda Chainitikun.

**Writing – review & editing:** Sudpreeda Chainitikun, Jose Rodrigo Espinosa Fernandez, Toshiaki Iwase, Kumiko Kida, Xiaoping Wang, Sadia Saleem, Bora Lim, Vicente Valero, Naoto T. Ueno.

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
