## [Decision Letter · Decision Letter 0]

15 Jan 2021

PONE-D-20-31752

Pathological complete response of adding targeted therapy to neoadjuvant chemotherapy for inflammatory breast cancer: a systematic review.

PLOS ONE

Dear Dr. Ueno,

Thank you for submitting your manuscript to PLOS ONE. After careful consideration, we feel that it has merit but does not fully meet PLOS ONE’s publication criteria as it currently stands. Therefore, we invite you to submit a revised version of the manuscript that addresses the points raised during the review process.

We would like you to revise the manuscript thematically and statistically as summarized below in the reviewers’ comments. 

We look forward to receiving your revised manuscript.

Kind regards,

Sudeep Gupta, M.D.

Academic Editor

PLOS ONE

Journal Requirements:

2. In your Methods section, please explain the reasons why the literature search was limited to 1998 and no earlier.

3. To comply with PLOS ONE submission guidelines, in your Methods section, please provide additional information regarding your statistical analyses. For more information on PLOS ONE's expectations for statistical reporting, please see https://journals.plos.org/plosone/s/submission-guidelines.#loc-statistical-reporting.

5. Please note that in order to use the direct billing option the corresponding author must be affiliated with the chosen institute. Please either amend your manuscript to change the affiliation or corresponding author, or email us at plosone@plos.org with a request to remove this option.

Additional Editor Comments:

The manuscript would be a useful piece of collated evidence on the theme but needs major revisions as suggested by the two reviewers.

Reviewers' comments:

Reviewer's Responses to Questions

**Comments to the Author**

1. Is the manuscript technically sound, and do the data support the conclusions?

Reviewer #1: Partly

Reviewer #2: No

2. Has the statistical analysis been performed appropriately and rigorously? 

Reviewer #1: Yes

Reviewer #2: Yes

3. Have the authors made all data underlying the findings in their manuscript fully available?

Reviewer #1: Yes

Reviewer #2: Yes

4. Is the manuscript presented in an intelligible fashion and written in standard English?

Reviewer #1: Yes

Reviewer #2: Yes

5. Review Comments to the Author

Reviewer #1: This paper reports on the pathological complete response rates with addition of targeted therapy to neoadjuvant chemotherapy for inflammatory breast cancer. It is an extensive review on a rare subtype of breast cancer. However, I have following concerns -

Methods

Outcome measure –

The main outcome of study is pathological complete response (path CR). However, the authors have considered two definitions of path CR - “the proportion of patients with absence of invasive cancer cells from breast or breast and axillary lymph nodes after neoadjuvant systemic treatment.’’

The current standard definition of path CR is absence of cancer cells from both breast and axillary Lymph nodes. Why have the authors taken the old definition of path CR, which is likely to overestimate the path CR rates?

Data Synthesis and analysis

High dose chemotherapy and hematopoietic stem cell support is not the current standard of care for breast cancer. The definition of path CR in high dose chemotherapy group is not as per the current standard, at best these studies overestimates the benefit of chemotherapy and the path CR rates. Moreover no targeted therapies were used in this group of trials. Since the aim of the present systemic review is to look at the improvement in path CR rates with addition of targeted therapies in inflammatory breast cancer patients (IBC), it would be better to drop the high dose chemotherapy group from the analysis so as to have a more homogenous comparator. Previous systemic review by Kim T et al (Clin Breast Cancer. 2006) has already addressed the path CR rates and issues with high dose chemotherapy in inflammatory breast cancer.

Study characteristics

Patients with locally recurrent IBC were included in the analysis. Since locally recurrent breast cancer have different disease biology and may not have the same response rates as in treatment naïve patients, their inclusion might have underestimated the path CR rates in both comparator and intervention arm. Request comment from authors on the same.

Results:

Pathological CR rates depend upon the breast cancer subtypes (Her 2 positive, TNBC and hormone receptor positive) and on the type of systemic therapies given. In this review all subtypes were clubbed together particularly in the comparator arm (chemotherapy alone group), which could have underestimated the path CR rates in this group. It would be better if the path CR rates can be given according to the breast cancer subtypes.

Similarly the weighted averages pCR rates are reported for all targeted therapies as a single group, where in panitumumab and tipifarnib are for Her2 negative breast cancer and trastuzumab is for Her2 positive breast cancer. It would be better to report the path CR rates with targeted therapies for each breast cancer subtype separately rather than as a single group.

Discussion:

Para 1 - “There was a difference of 18.6 percentage points between the weighted-average pCR for chemotherapy alone and the weighted-average pCR for targeted therapy plus chemotherapy”, this is probably an overstatement by authors since the weighted averages pCR rates are not as per the breast cancer subtypes. Both the targeted therapy and chemotherapy alone groups are heterogeneous and it is not correct to compare the biologically different subtypes directly. Would request a comment on this.

Para 4, 5 - Given the lack of benefit of bevacizumab in any breast cancer subtype, putting much emphasis on the role of bevacizumab in discussion (2 paragraphs have been written on this) seems excessive. Simply stating that addition of bevacizumab did not improve path CR in any breast cancer subtype would suffice.

Para 6 – weighted average path CR rates were calculated for non IBC patients as well. The methods section did not mention about this for non -IBC patients, moreover the primary aim of review is to analyze the path CR rates in IBC patients. Also, the search criteria included inflammatory breast cancer only; hence the calculation of weighted average path CR in non IBC patients from highly select group of trials may not be the ideal way and will not represent the true path CR rates in non IBC patients.

Para 9 – the schedule of chemotherapy and dose intensity of chemotherapy affect the path CR rates, which has not been analyzed in present study. Notably, 2 trials looked at efficacy of dose dense chemotherapy versus standard dose in IBC; both these trials have been included (ref 20, 35) but not analyzed from this point of view.

Reviewer #2: The authors have carried out and presented the results of a systematic review to determine whether neoadjuvant chemotherapy plus targeted therapy results in a higher pathologic complete response (pCR) rate than neoadjuvant chemotherapy alone in patients with IBC.

Comments:

1. The objective of this review is to compare pCR rates of NACT plus targeted versus NACT alone. The authors mention in the eligibility criteria of the review that study should evaluate neoadjuvant systemic treatment including targeted therapy before definitive surgery as an intervention and neoadjuvant systemic treatment not including targeted therapy before definitive surgery as a comparator which implies that RCTs comparing targeted versus non targeted therapies should be considered for the review. There is no comparator in this review and most of the studies are retrospective studies or trials comparing chemotherapy regimens without targeted therapy. Authors may consider removing the comparator from the eligibility criteria.

2. The authors should clearly mention if random or fixed effects model was used to derive the pooled estimate of pCR and software used for the analysis should also be described clearly.

3. A total of 24 studies has been considered for this review and the flow chart in fig 1 mentions only 23 studies which needs to be corrected.

4. Forest plot should be represented with heterogeneity statistic along with p value. A subgroup pooled estimate along with 95% CI can be show in the plot itself.

5. Chemotherapy with targeted therapy includes 329 patients from 9 studies. However the breakup in the table below fig 2 shows 105/9 and 168/4 which adds to only 274 patients from 9 studies. This discrepancy with respect to number of IBC patients/studies needs to be corrected.

6. Toxicities for all three subgroups can also be synthesized and presented along with a summary estimate and 95% CI.

6. PLOS authors have the option to publish the peer review history of their article (what does this mean?). If published, this will include your full peer review and any attached files.

Reviewer #1: No

Reviewer #2: No

---

## [Author Response · Author response to Decision Letter 0]

26 Feb 2021

Reviewer comments 

This paper reports on the pathological complete response rates with the addition of targeted therapy to neoadjuvant chemotherapy for inflammatory breast cancer. It is an extensive review of a rare subtype of breast cancer. However, I have the following concerns -

Methods 

Outcome measure – 

The main outcome of the study is pathological complete response (pCR). However, the authors have considered two definitions of pCR - “the proportion of patients with absence of invasive cancer cells from breast or breast and axillary lymph nodes after neoadjuvant systemic treatment.’’ The current standard definition of pCR is the absence of cancer cells from both breast and axillary Lymph nodes. Why have the authors taken the old definition of pCR, which is likely to overestimate the pCR rates?

-We agree with the reviewer and addressed the definition of pCR in our Discussion section, paragraph 7. In particular, we noted that in 3 high-dose chemotherapy studies, the old definition of pCR was used, which may have led to overestimation of pCR rates. But this would not change our principal finding that chemotherapy with targeted therapy was associated with the highest pCR rate. The studies of chemotherapy with targeted therapy and chemotherapy alone used the same recent pCR definition, which is why we analyzed these studies separately from the high-dose chemotherapy studies. Unfortunately, we have no way to re-classify the pCR data according to the new pCR definition in the 3 studies in which the old definition was used.

Data Synthesis and analysis 

High-dose chemotherapy and hematopoietic stem cell support are not the current standard of care for breast cancer. The definition of pCR in high dose chemotherapy group is not as per the current standard; at best, these studies overestimate the benefit of chemotherapy and the pCR rates. Moreover, no targeted therapies were used in this group of trials. Since the aim of the present systemic review is to look at the improvement in pCR rates with the addition of targeted therapies in inflammatory breast cancer patients (IBC), it would be better to drop the high dose chemotherapy group from the analysis to have a more homogenous comparator. Previous systemic review by Kim T et al. (Clin Breast Cancer. 2006) has already addressed the pCR rates and high dose chemotherapy issues in inflammatory breast cancer. 

-Thank you for your suggestions. We agree with your comments. Because some studies of high-dose chemotherapy used the old definition of pCR, removing the high-dose chemotherapy studies from the analysis would produce a more homogeneous comparison group. However, some readers might wonder about the pCR rates for chemotherapy plus targeted therapy vs. high-dose chemotherapy, which is why we retained the high-dose chemotherapy studies in our analysis.

-Regarding the Kim et al. study, as you mentioned, high-dose chemotherapy in that study generated a very high pCR rate of 32%. However, the study did not compare the pCR rate of high-dose chemotherapy with the pCR rate of chemotherapy plus targeted therapy. Our study separately analyzed pCR rates in 3 different groups for easier comparison and to minimize the effect of different pCR definitions. In our study, the pCR rates of chemotherapy with targeted therapy and chemotherapy alone were comparable based on the same recent pCR definition. The pCR rate for high-dose chemotherapy was overestimated with the old definition but still lower than the pCR rate of chemotherapy with targeted therapy. These findings do not change our conclusion that chemotherapy with targeted therapy was associated with the highest pCR rate.

Study characteristics 

Patients with locally recurrent IBC were included in the analysis. Since locally recurrent breast cancer has different disease biology and may not have the same response rates as in treatment naïve patients, their inclusion might have underestimated the pCR rates in both comparator and intervention arm. Request comment from authors on the same. 

-We agree with your comments. Inclusion criteria in the Palazzo et al. study allowed for both stage III and locally recurrent IBC. All 34 patients include in our analysis had stage III IBC. We have added this information in a footnote to Table 1.

Results:

Pathological CR rates depend upon the breast cancer subtypes (Her2 positive, TNBC, and hormone receptor-positive) and on the type of systemic therapies given. In this review, all subtypes were clubbed together, particularly in the comparator arm (chemotherapy alone group), which could have underestimated the pCR rates in this group. It would be better if the pCR rates can be given according to the breast cancer subtypes. 

Similarly, the weighted averages pCR rates are reported for all targeted therapies as a single group. Panitumumab and tipifarnib are for Her2 negative breast cancer, and trastuzumab is for Her2 positive breast cancer. It would be better to report the pCR rates with targeted therapies for each breast cancer subtype separately rather than as a single group. 

-We agree that the breast cancer subtype impacts the pCR rate. The studies in our systematic review had heterogeneity in the populations. Some studies did not report the number of patients with each breast cancer subtype, as shown in the table of baseline patient characteristics (Table 1); in addition, some studies did not report pCR rate according to subtype. The studies of chemotherapy before 2001 did not report the number of HER2-positive patients because HER2 testing was not available. As was the case for the targeted therapy studies, most of these studies did not report the pCR rate for each subtype. Given these limitations, we analyzed and reported the average pCR rate only for the studies of chemotherapy plus anti-HER2 therapy in patients with HER2-positive IBC, which was the most homogeneous population and also applicable to real-word practice. The pCR rates of the other IBC subtypes were not feasible because of the nature of the dataset.

Discussion:

Para 1 - “There was a difference of 18.6 percentage points between the weighted-average pCR for chemotherapy alone and the weighted-average pCR for targeted therapy plus chemotherapy”, this is probably an overstatement by authors since the weighted averages pCR rates are not as per the breast cancer subtypes. Both the targeted therapy and chemotherapy alone groups are heterogeneous, and it is not correct to compare the biologically different subtypes directly. Would request a comment on this.

-We edited to clarify as follows: 

“There was a difference of 18.6 percentage points between the weighted-average pCR for chemotherapy alone and the weighted-average pCR for targeted therapy plus chemotherapy when we analyzed all patients with IBC regardless of subtype. Anti-HER2 targeted therapy showed promising results for patients with HER2-positive IBC treated with chemotherapy plus anti-HER2 therapy, with a weighted-average pCR rate of 54.3%.”

Para 4, 5 - Given the lack of benefit of bevacizumab in any breast cancer subtype, putting much emphasis on the role of bevacizumab in discussion (2 paragraphs have been written on this) seems excessive. Simply stating that addition of bevacizumab did not improve pCR in any breast cancer subtype would suffice. 

-We shortened both paragraphs as suggested in your comments.

Para 6 – weighted average pCR rates were calculated for non-IBC patients as well. The methods section did not mention about this for non -IBC patients, moreover the primary aim of review is to analyze the pCR rates in IBC patients. Also, the search criteria included inflammatory breast cancer only; hence the calculation of weighted average pCR in non-IBC patients from highly select group of trials may not be the ideal way and will not represent the true pCR rates in non-IBC patients. 

-We totally agree with your comment. This issue is addressed in the following sentence in the Discussion section, paragraph 6: “In our systematic review, results for non-IBC patients cannot be directly compared to results for IBC patients because the search strategy was initiated on a research question that focused on IBC patients, and the study was not designed for non-IBC patients.” In response to your comment, we have added the following sentence for emphasis: “Hence, the non-IBC results in our systematic review were from highly selected patients and may not be representative of results in the overall population of patients with non-IBC.”

Para 9 – the schedule of chemotherapy and dose intensity of chemotherapy affect the pCR rates, which has not been analyzed in present study. Notably, 2 trials looked at efficacy of dose dense chemotherapy versus standard dose in IBC; both these trials have been included (ref 20, 35) but not analyzed from this point of view.

-We agree that this is a limitation, and we have discussed it in the limitations paragraph of the Discussion section, sentence 2. We agreed that dose-intensity is related to pCR rates, but it was not feasible to take into account dose-intensity in our data analysis.

Reviewer #2: The authors have carried out and presented the results of a systematic review to determine whether neoadjuvant chemotherapy plus targeted therapy results in a higher pathologic complete response (pCR) rate than neoadjuvant chemotherapy alone in patients with IBC.

1. The objective of this review is to compare pCR rates of NACT plus targeted versus NACT alone. The authors mention in the eligibility criteria of the review that study should evaluate neoadjuvant systemic treatment including targeted therapy before definitive surgery as an intervention and neoadjuvant systemic treatment not including targeted therapy before definitive surgery as a comparator which implies that RCTs comparing targeted versus non targeted therapies should be considered for the review. There is no comparator in this review and most of the studies are retrospective studies or trials comparing chemotherapy regimens without targeted therapy. Authors may consider removing the comparator from the eligibility criteria.

-We edited the Study Selection subsection of the Methods section following your suggestions.

2. The authors should clearly mention if random or fixed effects model was used to derive the pooled estimate of pCR and software used for the analysis should also be described clearly.

-As indicated in the last paragraph of the Methods section, we calculated a weighted-average pCR rate, defined as the total number of patients achieving pCR across all studies in the group divided by the total number of patients across all studies in the group. Confidence intervals for proportions were computed using the method of Clopper and Pearson. We did not use a pooled data analysis by fixed or random effect model because almost all of the studies we identified in our search were single-arm studies without a comparison group. We could not move forward to the meta-analysis process because the heterogeneity of the single-arm studies would have necessitated special statistical techniques to assemble a suitable comparator group. Such an approach would have yielded unreliable results, which is why we did not proceed to a formal meta-analysis

3. A total of 24 studies has been considered for this review and the flow chart in fig 1 mentions only 23 studies which needs to be corrected.

-Our systematic review had 23 studies. The study by Nahleh et al had both a chemotherapy-with-targeted therapy arm and a chemotherapy-alone arm. Therefore, results from this study are reported in both parts of Table 1.

4. Forest plot should be represented with heterogeneity statistic along with p value. A subgroup pooled estimate along with 95% CI can be show in the plot itself.

-We added the 95% CIs in the plot. 

5. Chemotherapy with targeted therapy includes 329 patients from 9 studies. However the breakup in the table below fig 2 shows 105/9 and 168/4 which adds to only 274 patients from 9 studies. This discrepancy with respect to number of IBC patients/studies needs to be corrected.

-We corrected the numbers in Figure 2.

6. Toxicities for all three subgroups can also be synthesized and presented along with a summary estimate and 95% CI.

-We found that the toxic effects were totally different in the different subgroups. The toxic effects of chemotherapy were neutropenia, infection, etc. The toxic effects of targeted therapy were decline of LVEF and headache for trastuzumab and rash and diarrhea for panitumumab. Synthesis of these toxicity by numerical incidence rate may not appropriate. Even though some toxic effects were rare but very serious, such as decline of LVEF.

---

## [Editor Report · Decision Letter 1]

1 Mar 2021

PONE-D-20-31752R1

Pathological complete response of adding targeted therapy to neoadjuvant chemotherapy for inflammatory breast cancer: a systematic review.

PLOS ONE

Dear Dr. Ueno,

Thank you for submitting your manuscript to PLOS ONE. After careful consideration, we feel that it has merit but does not fully meet PLOS ONE’s publication criteria as it currently stands. Therefore, we invite you to submit a revised version of the manuscript that addresses the points raised during the review process.

Please see the comments and action points below.

We look forward to receiving your revised manuscript.

Kind regards,

Sudeep Gupta, M.D.

Academic Editor

PLOS ONE

Additional Editor Comments (if provided):

Thanks for the revised manuscript. Although you have agreed with many of Reviewers comments you have made only minor changes to the manuscript. The revised manuscript has major limitations. Specifically please address the following, if you can, and resubmit. The action points are in CAPITALS.

Reviewer comments

This paper reports on the pathological complete response rates with the addition of targeted therapy to neoadjuvant chemotherapy for inflammatory breast cancer. It is an extensive review of a rare subtype of breast cancer. However, I have the following concerns -

Methods

Outcome measure –

The main outcome of the study is pathological complete response (pCR). However, the authors have considered two definitions of pCR - “the proportion of patients with absence of invasive cancer cells from breast or breast and axillary lymph nodes after neoadjuvant systemic treatment.’’ The current standard definition of pCR is the absence of cancer cells from both breast and axillary Lymph nodes. Why have the authors taken the old definition of pCR, which is likely to overestimate the pCR rates?

-We agree with the reviewer and addressed the definition of pCR in our Discussion section, paragraph 7. In particular, we noted that in 3 high-dose chemotherapy studies, the old definition of pCR was used, which may have led to overestimation of pCR rates. But this would not change our principal finding that chemotherapy with targeted therapy was associated with the highest pCR rate. The studies of chemotherapy with targeted therapy and chemotherapy alone used the same recent pCR definition, which is why we analyzed these studies separately from the high-dose chemotherapy studies. Unfortunately, we have no way to re-classify the pCR data according to the new pCR definition in the 3 studies in which the old definition was used. PLEASE INCLUDE THE DEFINITION OF pCR IN A SEPARATE COLUMN AGAINST EACH REGIMEN IN FIGURE 2.

Data Synthesis and analysis

High-dose chemotherapy and hematopoietic stem cell support are not the current standard of care for breast cancer. The definition of pCR in high dose chemotherapy group is not as per the current standard; at best, these studies overestimate the benefit of chemotherapy and the pCR rates. Moreover, no targeted therapies were used in this group of trials. Since the aim of the present systemic review is to look at the improvement in pCR rates with the addition of targeted therapies in inflammatory breast cancer patients (IBC), it would be better to drop the high dose chemotherapy group from the analysis to have a more homogenous comparator. Previous systemic review by Kim T et al. (Clin Breast Cancer. 2006) has already addressed the pCR rates and high dose chemotherapy issues in inflammatory breast cancer.

-Thank you for your suggestions. We agree with your comments. Because some studies of high-dose chemotherapy used the old definition of pCR, removing the high-dose chemotherapy studies from the analysis would produce a more homogeneous comparison group. However, some readers might wonder about the pCR rates for chemotherapy plus targeted therapy vs. high-dose chemotherapy, which is why we retained the high-dose chemotherapy studies in our analysis.

-Regarding the Kim et al. study, as you mentioned, high-dose chemotherapy in that study generated a very high pCR rate of 32%. However, the study did not compare the pCR rate of high-dose chemotherapy with the pCR rate of chemotherapy plus targeted therapy. Our study separately analyzed pCR rates in 3 different groups for easier comparison and to minimize the effect of different pCR definitions. In our study, the pCR rates of chemotherapy with targeted therapy and chemotherapy alone were comparable based on the same recent pCR definition. The pCR rate for high-dose chemotherapy was overestimated with the old definition but still lower than the pCR rate of chemotherapy with targeted therapy. These findings do not change our conclusion that chemotherapy with targeted therapy was associated with the highest pCR rate. PLEASE REMOVE THE STUDIES OF HIGH DOSE CHEMOTHERAPY FROM YOUR MAIN ANALYSIS INCLUDING FIGURE 2. YOU CAN INCLUDE THEM IN A SEPARATE FIGURE IN SUPPLEMENTARY MATERIAL.

Study characteristics

Patients with locally recurrent IBC were included in the analysis. Since locally recurrent breast cancer has different disease biology and may not have the same response rates as in treatment naïve patients, their inclusion might have underestimated the pCR rates in both comparator and intervention arm. Request comment from authors on the same.

-We agree with your comments. Inclusion criteria in the Palazzo et al. study allowed for both stage III and locally recurrent IBC. All 34 patients include in our analysis had stage III IBC. We have added this information in a footnote to Table 1. YOUR REPLY IS ACCEPTED

Results:

Pathological CR rates depend upon the breast cancer subtypes (Her2 positive, TNBC, and hormone receptor-positive) and on the type of systemic therapies given. In this review, all subtypes were clubbed together, particularly in the comparator arm (chemotherapy alone group), which could have underestimated the pCR rates in this group. It would be better if the pCR rates can be given according to the breast cancer subtypes.

Similarly, the weighted averages pCR rates are reported for all targeted therapies as a single group. Panitumumab and tipifarnib are for Her2 negative breast cancer, and trastuzumab is for Her2 positive breast cancer. It would be better to report the pCR rates with targeted therapies for each breast cancer subtype separately rather than as a single group.

-We agree that the breast cancer subtype impacts the pCR rate. The studies in our systematic review had heterogeneity in the populations. Some studies did not report the number of patients with each breast cancer subtype, as shown in the table of baseline patient characteristics (Table 1); in addition, some studies did not report pCR rate according to subtype. The studies of chemotherapy before 2001 did not report the number of HER2-positive patients because HER2 testing was not available. As was the case for the targeted therapy studies, most of these studies did not report the pCR rate for each subtype. Given these limitations, we analyzed and reported the average pCR rate only for the studies of chemotherapy plus anti-HER2 therapy in patients with HER2-positive IBC, which was the most homogeneous population and also applicable to real-word practice. The pCR rates of the other IBC subtypes were not feasible because of the nature of the dataset. YOUR REPLY IS ACCEPTED

Discussion:

Para 1 - “There was a difference of 18.6 percentage points between the weighted-average pCR for chemotherapy alone and the weighted-average pCR for targeted therapy plus chemotherapy”, this is probably an overstatement by authors since the weighted averages pCR rates are not as per the breast cancer subtypes. Both the targeted therapy and chemotherapy alone groups are heterogeneous, and it is not correct to compare the biologically different subtypes directly. Would request a comment on this.

-We edited to clarify as follows:

“There was a difference of 18.6 percentage points between the weighted-average pCR for chemotherapy alone and the weighted-average pCR for targeted therapy plus chemotherapy when we analyzed all patients with IBC regardless of subtype. Anti-HER2 targeted therapy showed promising results for patients with HER2-positive IBC treated with chemotherapy plus anti-HER2 therapy, with a weighted-average pCR rate of 54.3%.” YOUR REPLY IS ACCEPTED

Para 4, 5 - Given the lack of benefit of bevacizumab in any breast cancer subtype, putting much emphasis on the role of bevacizumab in discussion (2 paragraphs have been written on this) seems excessive. Simply stating that addition of bevacizumab did not improve pCR in any breast cancer subtype would suffice.

-We shortened both paragraphs as suggested in your comments. YOUR REPLY IS ACCEPTED

Para 6 – weighted average pCR rates were calculated for non-IBC patients as well. The methods section did not mention about this for non -IBC patients, moreover the primary aim of review is to analyze the pCR rates in IBC patients. Also, the search criteria included inflammatory breast cancer only; hence the calculation of weighted average pCR in non-IBC patients from highly select group of trials may not be the ideal way and will not represent the true pCR rates in non-IBC patients.

-We totally agree with your comment. This issue is addressed in the following sentence in the Discussion section, paragraph 6: “In our systematic review, results for non-IBC patients cannot be directly compared to results for IBC patients because the search strategy was initiated on a research question that focused on IBC patients, and the study was not designed for non-IBC patients.” In response to your comment, we have added the following sentence for emphasis: “Hence, the non-IBC results in our systematic review were from highly selected patients and may not be representative of results in the overall population of patients with non-IBC.” YOUR REPLY IS ACCEPTED

Para 9 – the schedule of chemotherapy and dose intensity of chemotherapy affect the pCR rates, which has not been analyzed in present study. Notably, 2 trials looked at efficacy of dose dense chemotherapy versus standard dose in IBC; both these trials have been included (ref 20, 35) but not analyzed from this point of view.

-We agree that this is a limitation, and we have discussed it in the limitations paragraph of the Discussion section, sentence 2. We agreed that dose-intensity is related to pCR rates, but it was not feasible to take into account dose-intensity in our data analysis. YOUR REPLY IS ACCEPTED

Reviewer #2: The authors have carried out and presented the results of a systematic review to determine whether neoadjuvant chemotherapy plus targeted therapy results in a higher pathologic complete response (pCR) rate than neoadjuvant chemotherapy alone in patients with IBC.

1. The objective of this review is to compare pCR rates of NACT plus targeted versus NACT alone. The authors mention in the eligibility criteria of the review that study should evaluate neoadjuvant systemic treatment including targeted therapy before definitive surgery as an intervention and neoadjuvant systemic treatment not including targeted therapy before definitive surgery as a comparator which implies that RCTs comparing targeted versus non targeted therapies should be considered for the review. There is no comparator in this review and most of the studies are retrospective studies or trials comparing chemotherapy regimens without targeted therapy. Authors may consider removing the comparator from the eligibility criteria.

-We edited the Study Selection subsection of the Methods section following your suggestions. YOUR REPLY IS ACCEPTED

2. The authors should clearly mention if random or fixed effects model was used to derive the pooled estimate of pCR and software used for the analysis should also be described clearly.

-As indicated in the last paragraph of the Methods section, we calculated a weighted-average pCR rate, defined as the total number of patients achieving pCR across all studies in the group divided by the total number of patients across all studies in the group. Confidence intervals for proportions were computed using the method of Clopper and Pearson. We did not use a pooled data analysis by fixed or random effect model because almost all of the studies we identified in our search were single-arm studies without a comparison group. We could not move forward to the meta-analysis process because the heterogeneity of the single-arm studies would have necessitated special statistical techniques to assemble a suitable comparator group. Such an approach would have yielded unreliable results, which is why we did not proceed to a formal meta-analysis. YOUR REPLY IS ACCEPTED

3. A total of 24 studies has been considered for this review and the flow chart in fig 1 mentions only 23 studies which needs to be corrected.

-Our systematic review had 23 studies. The study by Nahleh et al had both a chemotherapy-with-targeted therapy arm and a chemotherapy-alone arm. Therefore, results from this study are reported in both parts of Table 1. YOUR REPLY IS ACCEPTED

4. Forest plot should be represented with heterogeneity statistic along with p value. A subgroup pooled estimate along with 95% CI can be show in the plot itself.

-We added the 95% CIs in the plot. YOUR REPLY IS ACCEPTED

5. Chemotherapy with targeted therapy includes 329 patients from 9 studies. However the breakup in the table below fig 2 shows 105/9 and 168/4 which adds to only 274 patients from 9 studies. This discrepancy with respect to number of IBC patients/studies needs to be corrected.

-We corrected the numbers in Figure 2. YOUR REPLY IS ACCEPTED

6. Toxicities for all three subgroups can also be synthesized and presented along with a summary estimate and 95% CI.

-We found that the toxic effects were totally different in the different subgroups. The toxic effects of chemotherapy were neutropenia, infection, etc. The toxic effects of targeted therapy were decline of LVEF and headache for trastuzumab and rash and diarrhea for panitumumab. Synthesis of these toxicity by numerical incidence rate may not appropriate. Even though some toxic effects were rare but very serious, such as decline of LVEF. YOUR REPLY IS ACCEPTED

ADDITIONAL COMMENT FROM THE ACADEMIC EDITOR: TARGETED THERAPY IN NEOADJUVANT SETTING IN ADDITION TO CHEMOTHERAPY ESSENTIALLY APPLIES TO PATIENTS WITH HER2 POSITIVE DISEASE, IN WHOM IT IS ALREADY A STANDARD TREATMENT, WHETHER IBC OR NON-IBC. SO HOW ARE THE RESULTS OF THIS ANALYSIS USEFUL?

---

## [Author Response · Author response to Decision Letter 1]

29 Mar 2021

Reviewer comments and Response to reviewers

Thanks for the revised manuscript. Although you have agreed with many of Reviewers comments you have made only minor changes to the manuscript. The revised manuscript has major limitations. Specifically please address the following, if you can, and resubmit. The action points are in CAPITALS.

Reviewer comments

This paper reports on the pathological complete response rates with the addition of targeted therapy to neoadjuvant chemotherapy for inflammatory breast cancer. It is an extensive review of a rare subtype of breast cancer. However, I have the following concerns 

Methods

Outcome measure 

The main outcome of the study is pathological complete response (pCR). However, the authors have considered two definitions of pCR - “the proportion of patients with absence of invasive cancer cells from breast or breast and axillary lymph nodes after neoadjuvant systemic treatment.’’ The current standard definition of pCR is the absence of cancer cells from both breast and axillary Lymph nodes. Why have the authors taken the old definition of pCR, which is likely to overestimate the pCR rates?

-We agree with the reviewer and addressed the definition of pCR in our Discussion section, paragraph 7. In particular, we noted that in 3 high-dose chemotherapy studies, the old definition of pCR was used, which may have led to overestimation of pCR rates. But this would not change our principal finding that chemotherapy with targeted therapy was associated with the highest pCR rate. The studies of chemotherapy with targeted therapy and chemotherapy alone used the same recent pCR definition, which is why we analyzed these studies separately from the high-dose chemotherapy studies. Unfortunately, we have no way to re-classify the pCR data according to the new pCR definition in the 3 studies in which the old definition was used. 

PLEASE INCLUDE THE DEFINITION OF pCR IN A SEPARATE COLUMN AGAINST EACH REGIMEN IN FIGURE 2. 

-The pCR definition was added in last column of Figure 2.

Data Synthesis and analysis

High-dose chemotherapy and hematopoietic stem cell support are not the current standard of care for breast cancer. The definition of pCR in high dose chemotherapy group is not as per the current standard; at best, these studies overestimate the benefit of chemotherapy and the pCR rates. Moreover, no targeted therapies were used in this group of trials. Since the aim of the present systemic review is to look at the improvement in pCR rates with the addition of targeted therapies in inflammatory breast cancer patients (IBC), it would be better to drop the high dose chemotherapy group from the analysis to have a more homogenous comparator. Previous systemic review by Kim T et al. (Clin Breast Cancer. 2006) has already addressed the pCR rates and high dose chemotherapy issues in inflammatory breast cancer.

-Thank you for your suggestions. We agree with your comments. Because some studies of high-dose chemotherapy used the old definition of pCR, removing the high-dose chemotherapy studies from the analysis would produce a more homogeneous comparison group. However, some readers might wonder about the pCR rates for chemotherapy plus targeted therapy vs. high-dose chemotherapy, which is why we retained the high-dose chemotherapy studies in our analysis.

-Regarding the Kim et al. study, as you mentioned, high-dose chemotherapy in that study generated a very high pCR rate of 32%. However, the study did not compare the pCR rate of high-dose chemotherapy with the pCR rate of chemotherapy plus targeted therapy. Our study separately analyzed pCR rates in 3 different groups for easier comparison and to minimize the effect of different pCR definitions. In our study, the pCR rates of chemotherapy with targeted therapy and chemotherapy alone were comparable based on the same recent pCR definition. The pCR rate for high-dose chemotherapy was overestimated with the old definition but still lower than the pCR rate of chemotherapy with targeted therapy. These findings do not change our conclusion that chemotherapy with targeted therapy was associated with the highest pCR rate. 

PLEASE REMOVE THE STUDIES OF HIGH DOSE CHEMOTHERAPY FROM YOUR MAIN ANALYSIS INCLUDING FIGURE 2. YOU CAN INCLUDE THEM IN A SEPARATE FIGURE IN SUPPLEMENTARY MATERIAL. 

-We moved the high-dose chemotherapy data to Appendix section.

Study characteristics

Patients with locally recurrent IBC were included in the analysis. Since locally recurrent breast cancer has different disease biology and may not have the same response rates as in treatment naïve patients, their inclusion might have underestimated the pCR rates in both comparator and intervention arm. Request comment from authors on the same.

-We agree with your comments. Inclusion criteria in the Palazzo et al. study allowed for both stage III and locally recurrent IBC. All 34 patients include in our analysis had stage III IBC. We have added this information in a footnote to Table 1. 

YOUR REPLY IS ACCEPTED

Results:

Pathological CR rates depend upon the breast cancer subtypes (Her2 positive, TNBC, and hormone receptor-positive) and on the type of systemic therapies given. In this review, all subtypes were clubbed together, particularly in the comparator arm (chemotherapy alone group), which could have underestimated the pCR rates in this group. It would be better if the pCR rates can be given according to the breast cancer subtypes.

Similarly, the weighted averages pCR rates are reported for all targeted therapies as a single group. Panitumumab and tipifarnib are for Her2 negative breast cancer, and trastuzumab is for Her2 positive breast cancer. It would be better to report the pCR rates with targeted therapies for each breast cancer subtype separately rather than as a single group.

-We agree that the breast cancer subtype impacts the pCR rate. The studies in our systematic review had heterogeneity in the populations. Some studies did not report the number of patients with each breast cancer subtype, as shown in the table of baseline patient characteristics (Table 1); in addition, some studies did not report pCR rate according to subtype. The studies of chemotherapy before 2001 did not report the number of HER2-positive patients because HER2 testing was not available. As was the case for the targeted therapy studies, most of these studies did not report the pCR rate for each subtype. Given these limitations, we analyzed and reported the average pCR rate only for the studies of chemotherapy plus anti-HER2 therapy in patients with HER2-positive IBC, which was the most homogeneous population and also applicable to real-word practice. The pCR rates of the other IBC subtypes were not feasible because of the nature of the dataset. 

YOUR REPLY IS ACCEPTED

Discussion:

Para 1 - “There was a difference of 18.6 percentage points between the weighted-average pCR for chemotherapy alone and the weighted-average pCR for targeted therapy plus chemotherapy”, this is probably an overstatement by authors since the weighted averages pCR rates are not as per the breast cancer subtypes. Both the targeted therapy and chemotherapy alone groups are heterogeneous, and it is not correct to compare the biologically different subtypes directly. Would request a comment on this.

-We edited to clarify as follows:

“There was a difference of 18.6 percentage points between the weighted-average pCR for chemotherapy alone and the weighted-average pCR for targeted therapy plus chemotherapy when we analyzed all patients with IBC regardless of subtype. Anti-HER2 targeted therapy showed promising results for patients with HER2-positive IBC treated with chemotherapy plus anti-HER2 therapy, with a weighted-average pCR rate of 54.3%.” 

YOUR REPLY IS ACCEPTED

Para 4, 5 - Given the lack of benefit of bevacizumab in any breast cancer subtype, putting much emphasis on the role of bevacizumab in discussion (2 paragraphs have been written on this) seems excessive. Simply stating that addition of bevacizumab did not improve pCR in any breast cancer subtype would suffice.

-We shortened both paragraphs as suggested in your comments. 

YOUR REPLY IS ACCEPTED

Para 6 – weighted average pCR rates were calculated for non-IBC patients as well. The methods section did not mention about this for non -IBC patients, moreover the primary aim of review is to analyze the pCR rates in IBC patients. Also, the search criteria included inflammatory breast cancer only; hence the calculation of weighted average pCR in non-IBC patients from highly select group of trials may not be the ideal way and will not represent the true pCR rates in non-IBC patients.

-We totally agree with your comment. This issue is addressed in the following sentence in the Discussion section, paragraph 6: “In our systematic review, results for non-IBC patients cannot be directly compared to results for IBC patients because the search strategy was initiated on a research question that focused on IBC patients, and the study was not designed for non-IBC patients.” In response to your comment, we have added the following sentence for emphasis: “Hence, the non-IBC results in our systematic review were from highly selected patients and may not be representative of results in the overall population of patients with non-IBC.” 

YOUR REPLY IS ACCEPTED

Para 9 – the schedule of chemotherapy and dose intensity of chemotherapy affect the pCR rates, which has not been analyzed in present study. Notably, 2 trials looked at efficacy of dose dense chemotherapy versus standard dose in IBC; both these trials have been included (ref 20, 35) but not analyzed from this point of view.

-We agree that this is a limitation, and we have discussed it in the limitations paragraph of the Discussion section, sentence 2. We agreed that dose-intensity is related to pCR rates, but it was not feasible to take into account dose-intensity in our data analysis. YOUR REPLY IS ACCEPTED

Reviewer #2: The authors have carried out and presented the results of a systematic review to determine whether neoadjuvant chemotherapy plus targeted therapy results in a higher pathologic complete response (pCR) rate than neoadjuvant chemotherapy alone in patients with IBC.

1. The objective of this review is to compare pCR rates of NACT plus targeted versus NACT alone. The authors mention in the eligibility criteria of the review that study should evaluate neoadjuvant systemic treatment including targeted therapy before definitive surgery as an intervention and neoadjuvant systemic treatment not including targeted therapy before definitive surgery as a comparator which implies that RCTs comparing targeted versus non targeted therapies should be considered for the review. There is no comparator in this review and most of the studies are retrospective studies or trials comparing chemotherapy regimens without targeted therapy. Authors may consider removing the comparator from the eligibility criteria.

-We edited the Study Selection subsection of the Methods section following your suggestions. 

YOUR REPLY IS ACCEPTED.

2. The authors should clearly mention if random or fixed effects model was used to derive the pooled estimate of pCR and software used for the analysis should also be described clearly.

-As indicated in the last paragraph of the Methods section, we calculated a weighted-average pCR rate, defined as the total number of patients achieving pCR across all studies in the group divided by the total number of patients across all studies in the group. Confidence intervals for proportions were computed using the method of Clopper and Pearson. We did not use a pooled data analysis by fixed or random effect model because almost all of the studies we identified in our search were single-arm studies without a comparison group. We could not move forward to the meta-analysis process because the heterogeneity of the single-arm studies would have necessitated special statistical techniques to assemble a suitable comparator group. Such an approach would have yielded unreliable results, which is why we did not proceed to a formal meta-analysis. YOUR REPLY IS ACCEPTED

3. A total of 24 studies has been considered for this review and the flow chart in fig 1 mentions only 23 studies which needs to be corrected.

-Our systematic review had 23 studies. The study by Nahleh et al had both a chemotherapy-with-targeted therapy arm and a chemotherapy-alone arm. Therefore, results from this study are reported in both parts of Table 1. 

YOUR REPLY IS ACCEPTED

4. Forest plot should be represented with heterogeneity statistic along with p value. A subgroup pooled estimate along with 95% CI can be show in the plot itself.

-We added the 95% CIs in the plot. 

YOUR REPLY IS ACCEPTED

5. Chemotherapy with targeted therapy includes 329 patients from 9 studies. However the breakup in the table below fig 2 shows 105/9 and 168/4 which adds to only 274 patients from 9 studies. This discrepancy with respect to number of IBC patients/studies needs to be corrected.

-We corrected the numbers in Figure 2. 

YOUR REPLY IS ACCEPTED

6. Toxicities for all three subgroups can also be synthesized and presented along with a summary estimate and 95% CI.

-We found that the toxic effects were totally different in the different subgroups. The toxic effects of chemotherapy were neutropenia, infection, etc. The toxic effects of targeted therapy were decline of LVEF and headache for trastuzumab and rash and diarrhea for panitumumab. Synthesis of these toxicity by numerical incidence rate may not appropriate. Even though some toxic effects were rare but very serious, such as decline of LVEF. 

YOUR REPLY IS ACCEPTED

---

## [Editor Report · Decision Letter 2]

31 Mar 2021

Pathological complete response of adding targeted therapy to neoadjuvant chemotherapy for inflammatory breast cancer: a systematic review.

PONE-D-20-31752R2

Dear Dr. Ueno,

We are pleased to inform you that your manuscript has been judged scientifically suitable for publication and will be formally accepted for publication once it meets all outstanding technical requirements.

Kind regards,

Sudeep Gupta, M.D.

Academic Editor

PLOS ONE

Additional Editor Comments (optional):

Manuscript can be accepted for publication.
---

## [Editor Report · Acceptance letter]

7 Apr 2021

PONE-D-20-31752R2 

Pathological complete response of adding targeted therapy to neoadjuvant chemotherapy for inflammatory breast cancer: a systematic review 

Dear Dr. Ueno:

I'm pleased to inform you that your manuscript has been deemed suitable for publication in PLOS ONE. Congratulations! Your manuscript is now with our production department. 

Kind regards, 

on behalf of

Dr. Sudeep Gupta 

Academic Editor

PLOS ONE